# Impact of Aircraft Delays on Population Noise Exposure in Airport’s Surroundings

**DOI:** 10.3390/ijerph19158921

**Published:** 2022-07-22

**Authors:** Nermin Zijadić, Emir Ganić, Matija Bračić, Igor Štimac

**Affiliations:** 1International Airport Sarajevo, Kurta Schorka 36, 71126 Sarajevo, Bosnia and Herzegovina; nermin.zijadic@sarajevo-airport.ba; 2University of Belgrade, Faculty of Transport and Traffic Engineering, Vojvode Stepe 305, 11000 Belgrade, Serbia; 3University of Zagreb, Faculty of Transport and Traffic Sciences, Vukelićeva 4, 10000 Zagreb, Croatia; mbracic@fpz.unizg.hr (M.B.); istimac@fpz.unizg.hr (I.Š.); 4Zagreb Airport Ltd., Ulica Rudolfa Fizira 1, p.p. 102, 10410 Velika Gorica, Croatia

**Keywords:** airport environmental management, total airport management, airport noise contour, noise modelling, noise annoyance

## Abstract

The motivation behind this research was to analyse the consequences of aircraft operations’ delays on cumulative noise levels produced upon the neighbouring communities and to estimate the relative change in the number of people annoyed by aircraft noise. Many studies showed that residents’ reactions to abrupt changes in noise exposure were more intense compared to the anticipated ones. Aircraft delays may cause such abrupt changes in noise exposure by increasing the traffic in some periods compared to the scheduled traffic. The methodology applied includes noise contour development for two different scenarios for intervals where aircraft delays occur. Only delays connected with the Total Airport Management (TAM) were analysed, since such delays can be influenced by airports. The first scenario considered the influence of aircraft operations on population noise exposure without TAM delays, whereas the second one included all delayed flights (actual traffic). The proposed method was tested through case studies of three southeast European airports. The results showed that the highest potential of decrease in the number of people annoyed by the noise was recorded at Niš Airport (59%), followed by Zadar Airport (49%) and Sarajevo Airport (25%). Similar results were obtained in the context of highly annoyed people.

## 1. Introduction

Air traffic noise is one of the primary environmental problems faced by modern airports and will probably remain one of the limiting airport development factors in the future. Forecasts show that by 2039, air traffic volume in Europe is expected to increase about twice compared to 2019 [1]. Due to the continual growth of air traffic globally, the number of airports facing noise problems is increasing, i.e., the number of airports introducing specific noise management measures is also growing [2].

The first step in perceiving the magnitude of the noise problem is to assess the number of people exposed to aircraft noise by determining a noise contour. Furthermore, in order to determine the impact of aircraft noise on the population in airport surroundings, it is not enough to quantify the noise; instead, it is important to evaluate the noise effect through health risk, sleep, blood pressure and annoyance [3]. For that purpose, the European Commission developed a methodology to calculate the number of people highly annoyed and highly sleep disturbed by aircraft noise [4].

The latest estimations for population’s aircraft noise exposure according to the European Environment Agency [5] indicate that more than 3.2 million people are exposed to day–evening–night noise levels (L_den_) of 55 dB or higher. Although the L_den_ is a widely accepted noise indicator that considers long-term average sound level determined over all the day–evening–night periods of the year, it does not reflect the changes in noise levels on a short-term basis that could significantly differ due to the uneven number of operations, e.g., per hour.

Many studies have shown that residents’ reactions to abrupt changes in noise exposure are more intense compared to predicted responses from steady-state exposure–response relationships [6,7,8]. One of the reasons for such abrupt changes in noise exposure could be caused by aircraft delays, since in some periods of the day, the traffic could be significantly increased compared to the usual traffic to which the residents are accustomed.

Delays arise from aircraft operations and represent a negative side effect of the air transport system [9]. They can occur in any air transport subsystem and can significantly impact the environment in terms of aircraft noise and engine emissions, especially in the case of long delays. To analyse and classify the source of delays and interaction among all stakeholders, the International Air Transport Association (IATA) introduced the Airport Handling Manual (AHM) as a standard set of codes and descriptions of delays [10]. In this document, delays are grouped into several categories and only some of them are associated with airports. Airports can achieve a reduction in the effects of certain delays by applying the appropriate management, such as Airport Collaborative Decision Making (A-CDM) and the Total Airport Management (TAM) concept.

A-CDM refers to the process of data sharing, whereby airports, airlines, other stakeholders and the air navigation service provider (ANSP) share information to make operational decisions [11]. The concept of Total Airport Management (TAM) was originally introduced by DLR (Deutsches Zentrum für Luft und Raumfahrt—German Aerospace Center) and EUROCONTROL in 2006 [12] with the aim to improve cooperation between various airport stakeholders and ensure advanced, collaborative and coordinated planning of airport operations.

The subject of this research is aircraft operations and related delays, with focus on the consequences of these delays in terms of noise impact on the population around an airport. Only sub-delays connected with the Total Airport Management (passenger and cargo service and aircraft ramp handling) were analysed, since these can be influenced by airports. The purpose of this research is to demonstrate how TAM-related delays can impact the estimated noise exposure and annoyance of the population living in the vicinity of an airport.

In the following sections, an overview of the literature and methodology adopted in this paper are provided. In Section 4 of the article, the authors validate and demonstrate the proposed approach through a case study of three southeast European airports. Section 5 discusses the research results. Finally, there are concluding remarks and suggestions on how the application of this model may be taken forward.

## 2. Literature Review

A delay occurs if there is a particular deviation between the scheduled and actual time of aircraft operation [9]. The delay problem is analysed from different perspectives depending on whether the delay is considered by airports, ANSP providers or airlines. Delays in the air traffic system are, in most cases, difficult to predict and depend on different variables that are not necessarily correlated. To illustrate this, one of the variables could be bad meteorological conditions on the route or a sudden volcanic eruption with ash that will significantly change aircraft routes and cause significant delays. These examples are not correlated with, for example, passenger delays at the airport, when they forget their travel documents.

In many studies [13,14,15,16], predictive modelling is used to investigate the algorithms that most accurately predict delays. In general, limited airport capacity represents the main bottleneck of the overall air traffic system and generates a large proportion of delays. The airport capacity problem in terms of aircraft delay was researched through various aspects: slot adherence [17], flight rescheduling [18], congestion charges [19] and environmental efficiency [20]. From the air transport perspective, airlines have the most significant negative financial impact in the case of delays, which was researched in [21,22,23,24].

Based on the source, delays can be defined as primary and reactionary. Primary delays are manifested as process irregularities by one of the stakeholders, while reactionary delays (propagated delays) occur due to the interaction between different stakeholders involved in integrated processes: aircraft flight, air traffic control, ground handling and dispatch processes. Since flight delay propagation has a significant impact on air traffic performance, several research efforts have been conducted to better understand the delay patterns and to predict the effects of the delays on the air traffic network [25,26,27].

The air transport industry is also trying to mitigate the negative effect of propagated delays by introducing new operational concepts, such as A-CDM and TAM [28]. Airports need to evolve from today’s perspective where they behave as individual support tools and become components of an integrated airport information architecture to act as holistic decision–support tools for all airport partners [29]. The next step will be Performance-Based Airport Management (PBAM), developed by combining the existing concepts of TAM and Performance-Based Management (PBM). The main characteristic of PBAM is its specific focus on collaborative management according to Key Performance Indicators (KPIs) targeting economic, ecologic, safety, capacity and social demands [30,31].

It is apparent that there is a bundle of research that addresses the problem of delays at airports. The same can be concluded for the airport noise issue, as versatile research efforts are made discussing this problem [2,32,33], but also analysing different mitigation strategies and solutions to it [34,35,36]. In his doctoral dissertation, Zijadić [37] researched an airport operation and management system and made basic assumptions for correlating TAM-related delays and population exposure to noise. To the best of the authors’ knowledge, there are no other research studies that directly connect these two issues and analyse the consequences of aircraft operations’ delays on cumulative noise levels produced to the neighbouring communities.

An analogy in this context can be made between delays and seasonality, since airports may experience noise problems during the peak season due to increased traffic, in similar way as during the peak hour due to delays [38,39].

Even if the overall noise levels remain unchanged, people may react strongly to aircraft noise due to the altered flight distribution between the day and night or within daytime or night-time hours [40,41]. In some cases, such redistribution could increase the L_den_ noise levels if some flights are shifted from day to evening and/or night hours. Therefore, some research efforts focused on maintaining more equal noise levels through noise allocation tools that could effectively reduce peak levels [42].

Since it is evident that the number of aircraft noise events and the distribution of these events over time contribute to residential noise annoyance, more research on the impact of aircraft delays on population noise exposure is needed. Hence, this was the actual motivation behind the study presented in this paper.

## 3. Methodology and Methods

The methodology of this research is defined according to the following steps: data collection, data pre-processing, calculation of affected population and aircraft noise contour modelling, as illustrated in Figure 1. Each step is briefly explained below, followed by a description of scientific methods used in this research.

### 3.1. Data Collection

The first step involves data collection from Airport Operational Data Base (AODB) and Aeronautical Information Publication (AIP). To assess impact of delays, data regarding delay codes (arrival and departure) and aircraft delays (in minutes) were collected. In order to develop noise contours and calculate noise impact on the population in the airport’s surroundings, it is necessary to collect the following data: date and time of aircraft operation, aircraft type, origin and destination airport, type of operation, aircraft maximum take-off mass (MTOM), runway identifier, and departure and arrival routes.

These elements represent high-value inputs for specialised noise modelling software. They are combined inside the noise model with the already included default data, such as different types of engines per aircraft, engine thrust specifications per each flight segment, etc. Noise contours are developed by merging all these data.

The next step is to collect data on the population in the airport’s surroundings (number and location). In this research, QGIS was used to present and analyse population data. The population source was the Global Human Settlement Layer (GHSL) GHS population grid, derived from EUROSTAT census data (2011) and ESM R2016-100m cells [43].

### 3.2. Data Pre-Processing

In this paper, any deviation between the scheduled and actual time of arrival and departure aircraft operations shall be considered as a delay. Although every delay could affect short-term noise exposure around the airport in the same way, regardless of the reason that caused the delay, in this research, we focused only on aircraft delays associated with TAM, since airports could impact such delays. These delays codes are: 11–19 (Passenger and Baggage); 21–29 (Cargo and Mail); 31–39 (Aircraft and Ramp Handling); 51–58 (Damage to Aircraft and Electronic Data Processing (EDP)), and 85–89 (Airport and Governmental Authorities) [10]. Delays related to adverse weather conditions, aircraft defects, cabin crew shortage and similar reasons beyond airports’ influence are not considered in this study. Nevertheless, this methodology could be applied in other research where all types of delays could be included.

The next step is to isolate the time intervals (time slots) when these delays occur. There are three criteria for selecting relevant time intervals for further analysis: (1) duration of the time interval, (2) the overall number of operations within each interval and (3) the number of operations with TAM-related delays within the interval. The first criterion relating to “duration of the time interval” strongly depends on the airport business model, aircraft frequency and peaks during the days. Based on those parameters, 15, 30 and 60 min intervals can be used. After the first criterion is defined, the second criterion is based on the number of operations, where the minimum number of operations with delay should be two or more. To fulfil the third criterion, a minimum of one operation with TAM related delay in all the delays should occur in chosen interval to enable comparison between operations with and without TAM. All three criteria are arbitrary and depend on the airport’s delay characteristics and distribution of aircraft operations during the day and the number of frequencies. It should be emphasized that within this research we analysed airports with different business models. Therefore, for the main and the largest international airport in the country, which is the main hub and has aircraft operations continuously over the year, with some characteristic peaks during the day, we propose 15 min intervals as relevant. However, for a small seasonality airport that has traffic with a strong seasonality influence (tourism) and where deviations are from one aircraft per day during winter to over 50 during summer, we use 60 min intervals as relevant. The same 60 min intervals are used for airports that are secondary airports in the country, with a low level of traffic compared to their primary airports.

To compare the population noise exposure in situations with and without TAM-related delays, two different scenarios were analysed for each of these intervals. The first scenario includes only flights that would occur in the observed time interval if there were no TAM delays (actual traffic without TAM delays). The second scenario represents a real situation and considers all flights (actual traffic with TAM delays). This means that the second scenario includes all delayed flights, regardless of the cause of delay, while the first scenario does not include the delayed flights that could be influenced by the airport through the Total Airport Management concept.

### 3.3. Aircraft Noise Contour Modelling

The noise metric that needs to be calculated for each location is the $L_{Aeq,T}$ or the A-weighted equivalent continuous sound level determined over the time period T:(1)$$L_{Aeq,T}=10\cdot log_{10}\left(\frac{1}{T}\sum_{i=1}^{n}10^{\frac{SEL_{i}}{10}}\right),$$
where n is the number of aircraft operations during time period T and $SEL_{i}$ is the sound exposure level of aircraft i.

There are many different noise modelling tools that could calculate such noise levels, and for this research, the FAA’s Integrated Noise Model (INM) was used [44]. INM is a computer model used worldwide and designed for conducting various noise impact assessment studies in the vicinity of airports. Noise exposure is estimated through noise–power–distance (NPD) data considering various input data, such as air traffic data, specific operation mode, thrust setting and other environmental factors. The main outputs of INM are either noise contours for an area of interest or the noise level at pre-defined locations/coordinates. The INM’s core computation modules are compliant with many international standards documents including European Civil Aviation Conference (ECAC) Document 29, which represents a standard method for computing noise contours around civil airports [44].

For the purpose of visualising noise levels around the airport, noise contours were used as standard output in airport noise analysis. All illustrations of noise contours were created using the OpenStreetMap background and QGIS application.

### 3.4. Calculation of Affected Population

After calculation of $L_{Aeq,T}$ noise levels, the number of people exposed to those noise levels at each location for each selected interval was determined. To assess the expected annoyance and harmful effects of aircraft noise upon population, dose–effect relation was used concerning the relation between annoyance and noise levels for air traffic noise. The total number of people annoyed by aircraft noise (NPA) is estimated using the polynomial approximation in (2) as suggested by the European Commission [45], where $L_{Aeq,T}$ is used instead of the L_den_ noise levels:(2)$$NPA=\overset{m}{\underset{l=1}{\sum }}P_{l}\cdot \left(\left(8.588\cdot 10^{-6}\cdot {\left(L_{Aeq,T,l}-37\right)}^{3}+1.777\cdot 10^{-2}\cdot {\left(L_{Aeq,T,l}-37\right)}^{2}+1.221\cdot \left(L_{Aeq,T,l}-37\right)\right)/100\right)$$
where m is the number of locations l; $P_{l}$ is the population at each location l; and $L_{Aeq,T,l}$ is *A*-weighted equivalent continuous sound level during time period T calculated for location l.

The European Commission also gives the approximation for estimating the total number of people highly annoyed by aircraft noise (NPHA) as follows (3):(3)$$NPHA=\overset{m}{\underset{l=1}{\sum }}P_{l}\left(\left(-9.199\cdot 10^{-5}\cdot {\left(L_{Aeq,T,l}-42\right)}^{3}+3.932\cdot \;10^{-2}\cdot {\left(L_{Aeq,T,l}-42\right)}^{2}+0.2939\cdot \left(L_{Aeq,T,l}-42\right)\right)/100\right)$$

Equation (2) indicates that people are annoyed by aircraft noise only when the $L_{Aeq,T}$ values are higher than 37 dB (A), while people are highly annoyed if the $L_{Aeq,T}$ values are higher than 42 dB (A), as in (3).

The noise annoyance assessment of *NPA* and *NHPA* was used in this research solely for relative comparisons of population noise exposure during short-term intervals. Since it combines calculated noise exposure levels and population affected at different locations into a single indicator, such noise annoyance assessment is convenient for comparing the two scenarios. Therefore, it should be borne in mind that the calculated values are not intended to be used as an absolute indicator for the number of people annoyed by noise in this research.

### 3.5. Methods

Several scientific methods were used in the research, as follows. The descriptive method was applied throughout the paper, mainly where definitions and concepts were explained, in the background part to show the research overview, and where the case study airports were described. The comparison method was applied in the research where aircraft delay operations were evaluated at three airports. This method was primarily used when the results from the airport traffic database (AODB) and the calculated noise levels were compared for the two scenarios, “with TAM” or “without TAM”, and in the segment showing how aircraft delays influence the population noise exposure. The statistical method was used mostly during the airport database processing, when the traffic data were collected and analysed to determine aircraft delays connected with the Total Airport Management. Methods of analysis and synthesis were used according to the collected information and traffic data to define airport operations and the relationship between aircraft delay and noise levels. Graphical and tabular representation is used for the aircraft noise contours, analytical results for each airport, as well as the case study comparisons between “with TAM” and “without TAM” scenarios.

## 4. Case Study Overview

In order to validate and demonstrate the applicability of the proposed approach, a case study was carried out at three different types of airports, located in three different countries. These airports are Sarajevo Airport in Bosnia and Hercegovina, Zadar Airport in Croatia and Niš “Constantine the Great” Airport in Serbia. A brief description of each airport is given below. The data relating to the operation numbers were collected directly from each airport. We received the airport operational databases (AODB) and analysed their traffic, aircraft operations, aircraft types, delays and other data valuable for this research.

### 4.1. Sarajevo Airport Case Study

Sarajevo Airport (ICAO airport code: LQSA) is the main airport in Bosnia and Herzegovina, located 7 km from the City of Sarajevo (population: 413,593 inhabitants in Sarajevo Canton). The airport is declared as International Civil Aviation Organization (ICAO) code 4D and can be categorised as joint civil/military airport. The airport operates with one asphalt runway 11/29, its dimensions are 2600 m × 45 m, and it is connected with the apron with three asphalt taxiways. During the peak hour, the apron at Sarajevo Airport can handle seven aircraft with aerodrome code letter C. The passenger terminal has a declared capacity of 1 million passengers per year. It should be emphasized that the airport serves mostly Full-Service Carriers (FSC); however, low-cost carriers (LCC) are also present. In 2019, Sarajevo Airport handled a total of 1,143,680 passengers, which was the record number of passengers in the airport’s history.

For the Sarajevo Airport case study, 2015 was used, when the airport handled 11,107 aircraft operations consisting of 5553 departures and 5554 arrivals. On arrival, 3563 operations were delayed, which meant that more than 64% of all arrival operations were delayed. On departure, the situation was slightly better, since around 46% of all departure operations were delayed (2554 operations). The fleet mix consisted of 143 different aircraft types, with these seven aircraft types operating 65% of all flights: Boeing 737–800 (13%), de Havilland Dragon (13%), Airbus 319 (11%), LET L-410 (10%), Airbus 320 (7%), ATR 72 (7%) and Airbus 321 (4%). All flights were assigned to four departure routes and three arrival routes. Most of the flights delayed at arrival (90%) were performed during daytime (07–19 h), followed by 6% of flights during the evening (19–23 h), while there were 4% of night flights (23–07 h). Most of the flights delayed at departure (94%) were performed during daytime (07–19 h), followed by 4% of flights during the evening (19–23 h), while there were 2% of night flights (23–07 h).

### 4.2. Zadar Airport Case Study

Zadar Airport (ICAO airport code: LDZD) is located 10 km from the City of Zadar (population of the wider area is 168,031 inhabitants). Due to its specific location on the coast, the airport entirely depends on tourism and has a high seasonality impact. The airport is declared as ICAO code 4D and can be categorised as joint civil/military airport with a military base located in the northern part of the airport complex. Zadar Airport is the only airport in Croatia with two runways. The first runway 04/22 (civil) has dimensions of 2000 m × 45 m, while the second (military) runway 13/31 has dimensions of 2500 m × 45 m. Both runways are used for civil and military operations. The runways are connected with the apron via ten taxiways. The passenger terminal is located on the west part of the airport and has maximum capacity of 600,000 passengers. The majority of aircraft landing at Zadar Airport are aircraft with aerodrome code letter C (65% of total operations). Due to its high seasonality, LCCs and FSCs are significantly present, mostly during the summer. The record year by the number of passengers at Zadar Airport was the year 2019, when 801,347 passengers travelled through the airport, and this year was used for the case study.

Yearly 2019 traffic comprised 11,046 aircraft operations (6922 commercial and 4124 General Aviation and other flight types). For this research, only commercial operations were analysed. The military runway 13/31 handled 92.4% of all operations, while civil runway 04/22 was used for only 7.6% of operations. On arrival, 1097 operations were delayed, which meant that 32% of all arrival operations were delayed. Regarding departure operations, 14%, or 468 operations, were delayed. The fleet mix consisted of 19 different aircraft types that were assigned to 18 departure routes and 13 arrival routes. Four aircraft types operate 92% of Zadar Airport commercial traffic: Boeing 737–800 (41%), Dash8-Q400 (24%), Airbus A320 (14%) and Airbus A319 (12%). Most of the flights (61.4%) were performed during daytime (07–19 h), followed by 25.9% of flights during the evening (19–23 h), while there 12.7% made up night flights (23–07 h). Most of the flights delayed at arrival (62.3%) were performed during the daytime (07–19 h), followed by 30.7% of flights during the evening (19–23 h), while there were 6.9% of night flights (23–07 h). Regarding departure operations, most of the flights delayed at departure (66.1%) were performed during the daytime (07–19 h), followed by 26.8% of flights during the evening (19–23 h), while there were 7.1% of night flights (23–07 h).

### 4.3. Niš “Constantine the Great” Airport Case Study

Niš “Constantine the Great” Airport (ICAO airport code: LYNI) is the second-largest and the second-busiest airport in Serbia, after Belgrade Airport. This airport is located 4 km northwest of the City of Niš (population: 255,901 inhabitants). The airport is declared as ICAO code 4D and can be categorised as joint civil/military airport. The airport has one asphalt runway (2500 × 45 m) and one smaller grass runway (1700 × 50 m), both of which have 11/29 orientation. One taxiway connects the asphalt runway to the apron, which has four aircraft stands available for commercial operations. Traffic is mainly oriented towards FSC, LCC and charter operations. For this case study, the year 2019 was used, which was the passenger record year at Niš Airport, when 422,255 passengers travelled through the airport.

In 2019, the airport recorded 3932 aircraft operations, with 1963 departures and 1969 arrivals. The distribution of operations between runways was slightly in favour of runway 29, which handled 2122 operations (54%), while runway 11 was used for 1810 operations (46%). On arrival, 835 operations were delayed, which meant that 42.4% of all arrival operations were delayed. Regarding departure operations, 50.6%, or 994 operations, were delayed. Even though during 2019 there were 83 different aircraft types present at Niš Airport, in more than three-quarters of operations, only the following four aircraft types were used: Airbus 319 (26.6%), Airbus 320 (23.1%), Boeing 737–800 (22.8%) and Airbus 321 (4.1%). All operations were assigned to 12 departure routes and 17 arrival routes. Most of the operations (73.1%) were performed during daytime (06–18 h), followed by 21.5% of evening flights (18–22 h), while there were 5.3% of night flights (22–06 h). Most of the flights delayed at arrival (69.0%) were performed during the daytime (06–18 h), followed by 25.0% of flights during the evening (18–22 h), while there were 6.0% of night flights (22–06 h). Regarding departure operations, most of the flights delayed at departure (72.5%) were performed during the daytime (06–18 h), followed by 25.1% of flights during the evening (18–22 h), while there were 2.4% of night flights (22–06 h).

## 5. Results and Discussion

### 5.1. TAM Delays Analysis

After analysing AODB data for Sarajevo, Zadar and Niš airports and filtering those operations with TAM-related delays, the results showed a difference in the number of operations and schedule distribution of the aircraft operations during the day. Therefore, for Sarajevo airport, we used 15 min intervals, while for Zadar and Niš airports, the 60 min intervals proved to be relevant.

For Sarajevo Airport, 43 isolated intervals during 2015 were found to have had several delays, out of which at least one delay was related to the Total Airport Management (TAM). Out of this number, 23 periods had delays only in arrivals, 16 periods had delays only in departures, while there were 4 periods with delays involving both arrival and departure flights.

The analysis of flight operations at Zadar Airport for 2019 revealed that there were 20 such periods with at least one TAM-related delay. A similar situation was observed for Niš Airport, where there were 24 such periods.

### 5.2. Noise Exposure Analysis

For each of these three airports, the number of people annoyed (using Equation (2)) and highly annoyed (using Equation (3)) by aircraft noise was estimated separately for both scenarios (actual traffic with and without TAM-related delays) and for each selected interval. Summary results describing minimum, maximum, average, mean and quartile values are presented in Table 1, while Appendix A contains more detailed data for each interval and airport.

The maximum number of annoyed people during one interval with TAM delays was 53,136, as observed around Sarajevo Airport. These numbers were drastically lower in the case of Zadar Airport and Niš Airport, where the maximum values were 869 and 17,836, respectively.

As mentioned in Section 3.4, these calculated values of NPA and NPHA are not intended to be used as absolute values, since they do not refer to the whole year (only short-term intervals) and are not based on the L_den_ noise indicator. The purpose is to use them solely for relative comparisons of population noise exposure between the two scenarios. Therefore, the magnitude of the impact of TAM-related delays on the population noise exposure in airport surroundings was observed through comparison of relative changes in the number of annoyed/highly annoyed people.

The first step is to determine the absolute change between the two scenarios. Secondly, to determine the relative change, the second scenario that considers all flights (actual traffic with TAM delays) is set as the reference point. Finally, the absolute difference is divided by the reference value to calculate the relative change. Since, in this case, the absolute changes are always smaller than the reference value, this change symbolises a relative decrease in the population affected by aircraft noise that could be achieved if the TAM-related delays were eliminated. The relative change is calculated for each individual interval and the results on an airport level are expressed as the average value of these changes per each interval.

The results for Sarajevo Airport indicate that the number of annoyed people could decrease by 25% on average and the number of highly annoyed people could decrease by 30% if TAM delays were eliminated. This decrease is even higher in the case of Zadar Airport and amounts to 49% and 58% for the number of annoyed people and the number of highly annoyed people, respectively. In the case of Niš Airport results, the magnitude of people who are exposed to aircraft noise surpasses the two previously mentioned airports. When TAM delays do not occur at Niš Airport, the estimated number of annoyed people decreases by 59%, while the estimated number of highly annoyed people decreases by 60%, on average.

A more detailed analysis for each interval and each airport is shown in Figure 2 (for NPA) and Figure 3 (for NPHA). The relative relationship between the two scenarios (with TAM and without TAM), for each interval separately, clearly reveals that the differences are not equally distributed. This is due to an uneven number of delayed flights within intervals. In the case of Sarajevo Airport, the highest relative decrease of 94% was recorded for the first interval, although the highest absolute difference was for the fourth observed interval, where the NPA could decrease from 46,508 to 8708 if there had been no TAM-related delays. For Zadar Airport, the relative decrease was the highest (100%) for the first three observed intervals, where the elimination of TAM delays could reduce the number of people annoyed by noise to zero. In the case of Niš Airport, the highest relative decrease was most noticeable in the third interval, where NPA decreased from 3022 to 224 (93% decrease). The relative difference in the number of people highly annoyed by aircraft noise between the two scenarios for each interval separately is also not equally distributed, as shown in Figure 3. For example, the values of relative decrease vary from 2% up to 98% reduction in NPHA for Sarajevo Airport, from 0% to 100% for Zadar Airport and from 4% to 94% for Niš Airport. Furthermore, for most of the intervals, the relative decrease for NPHA is higher than the one for the NPA, except for several cases for Zadar and Niš airports where the relative decrease is higher for NPA. For example, for the 13th observed interval at Niš Airport, the NPA is estimated to decrease by 73% (from 2890 to 772), while for the same interval, the NPHA is estimated to decrease by 58% (from 568 to 237).

### 5.3. Comparison of Noise Contours

In order to visualise the results more clearly, several intervals with pronounced differences in the number of people annoyed by noise between the scenarios with and without TAM-related delays were selected and the calculated noise contours for each airport are presented herein.

Figure 4 shows the *L_Aeq_*_,15min_ noise contours for Sarajevo Airport for one fifteen-minute period on 3 August 2015 (the fourth observed interval in Figure 2). The first noise contour (Figure 4a) represents the real situation (actual traffic with TAM-related delays) where within the 15 min interval, two Boeing 737–800 departed from runway 29 (heading northwest). At the same time, there were also two arrivals, Airbus 319 and Airbus 321, to runway 11 (approaching from the northwest). The second noise contour (Figure 4b) reflects the scenario in which only operations without TAM-related delays occur.

In this case, only two landings were involved, since two delayed departure operations were not supposed to occur within this interval. If the TAM-related measures had been implemented, these delays could have been obviated, thus, reducing the population noise exposure during the observed interval.

Figure 5 shows the *L_Aeq_*_,1h_ noise contours for Zadar Airport for a one-hour interval on 16 October 2019. In this example, the scenario with TAM-related delays (Figure 5a) contains two departures of Boeing 737-800 and Airbus 320 from runway 31 (heading northwest). However, since the Boeing 737-800 entered this interval only due to the delay, a scenario without TAM-related delays (Figure 5b) illustrates only the departure of Airbus 320.

The difference between the two scenarios is evidently visible, resulting in an increased number of people exposed to aircraft noise, most of who are located northwest of the airport in the direction of runway 31.

In the case of Niš Airport, the interval presented in Figure 6a illustrates one departure of Boeing 737-800 from runway 29 (heading northwest) and one arrival of Airbus A319 to runway 11 (approaching from the northwest). According to the flight plan, only one arrival operation was scheduled within this interval (Figure 6b). Since the departure was delayed, it affected the noise exposure of the population located northwest of the airport. All presented noise contours clearly indicate the magnitude of TAM-related delays’ influence on the population noise exposure and annoyance.

## 6. Conclusions

This paper evaluates the impact of aircraft delays on population noise exposure around an airport, emphasising the relationship between TAM-related delays and noise levels, because the airport can affect and reduce or eliminate such delays. In this paper, the authors proposed a new methodology to calculate the effect of noise exposure. Namely, specific time intervals to process flights with and without TAM-related delays were applied. In further steps, noise contours for two different scenarios were developed. In the first scenario, noise contours were developed based on the actual traffic without TAM-related delays. In the second scenario, the actual traffic with TAM-related delays was used to construct noise contours. To highlight the impact of TAM-related delays on the population in the airport surroundings, the European Commission’s methodology to determine the number of annoyed and highly annoyed people was applied through relative change comparison. This new approach emphasises the importance of aircraft delays on population noise exposure.

The case study was carried out for the airports of Sarajevo, Zadar and Niš, which all differ according to the manoeuvring area/airspace configuration, number of aircraft operations and type of air carriers. Two types of indicators used to analyse the results of this research are: the number of annoyed and highly annoyed people. If we compare flights with and without TAM delays, the highest average decrease in the number of annoyed people could be achieved at Niš Airport (59% reduction), followed by Zadar Airport (49%) and Sarajevo Airport (25%). In the context of highly annoyed people, the results are similar. The relative decrease in the number of highly annoyed people is most significant for Niš Airport (60%), followed by Zadar Airport (58%) and Sarajevo Airport (30%). These results show differences, mainly because of the airport layout, seasonality of air traffic together with the type and frequency of operating aircraft.

The conducted research and the model could be applied to any airport, with an adjustment in research parameters for selecting appropriate time intervals. The results of the calculated population noise exposure and presented noise maps clearly demonstrate the impact of TAM-related delays on noise increase.

In future research, it is recommended to apply this model to airports with a high level of traffic and different airport and terrain configuration. The example of Sarajevo Airport indicates that a significant effect in terms of a population noise exposure reduction can be obtained for airports located in the city or in the vicinity of largely populated areas.

Finally, it may be concluded that it is pivotal to reduce TAM-related delays, thus, decreasing noise annoyance in the population in the airport’s surroundings. This can be accomplished with the implementation of new technologies, such as Collaborative Decision Making and Total Airport Management Suite.

## Figures and Tables

**Figure 1 ijerph-19-08921-f001:**
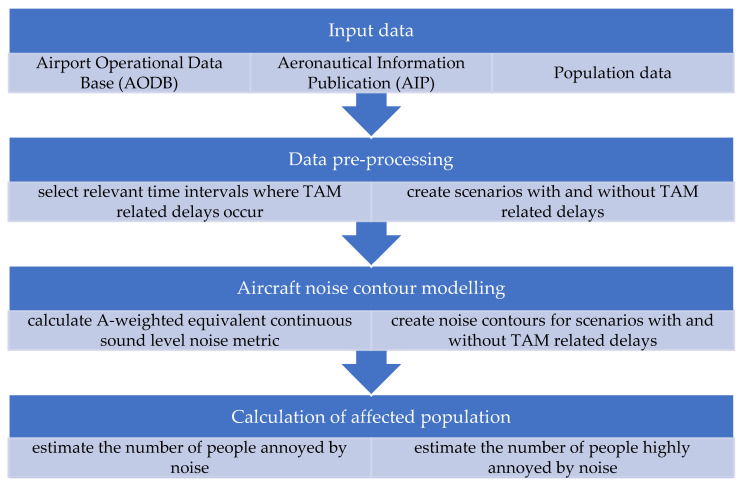
Research methodology.

**Figure 2 ijerph-19-08921-f002:**
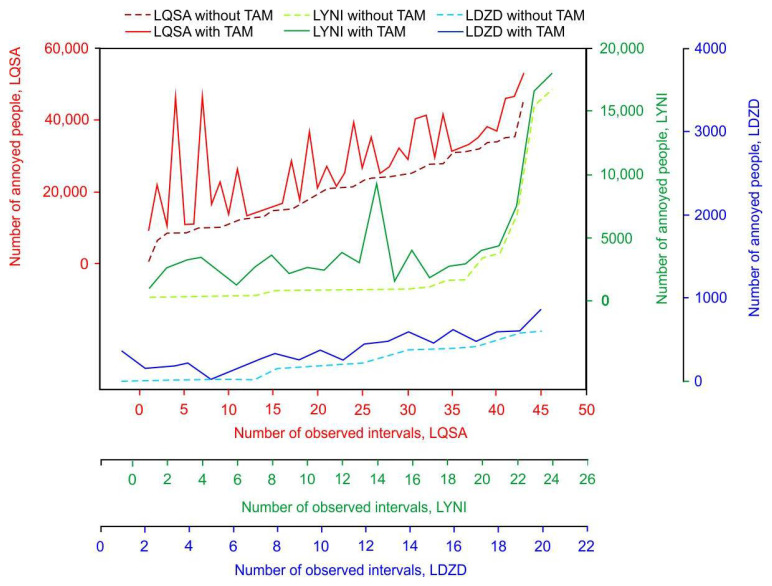
Comparative analysis of annoyed people around airports of Sarajevo, Zadar and Niš.

**Figure 3 ijerph-19-08921-f003:**
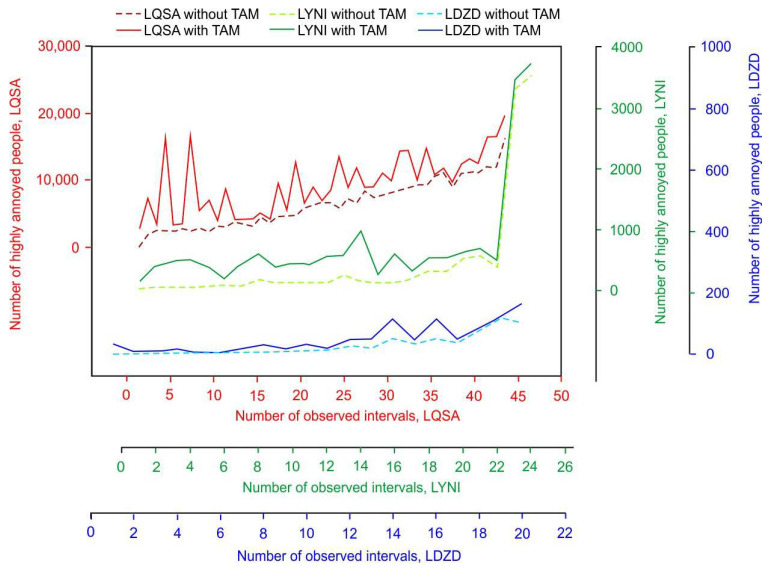
Comparative analysis of highly annoyed people around airports of Sarajevo, Zadar and Niš.

**Figure 4 ijerph-19-08921-f004:**
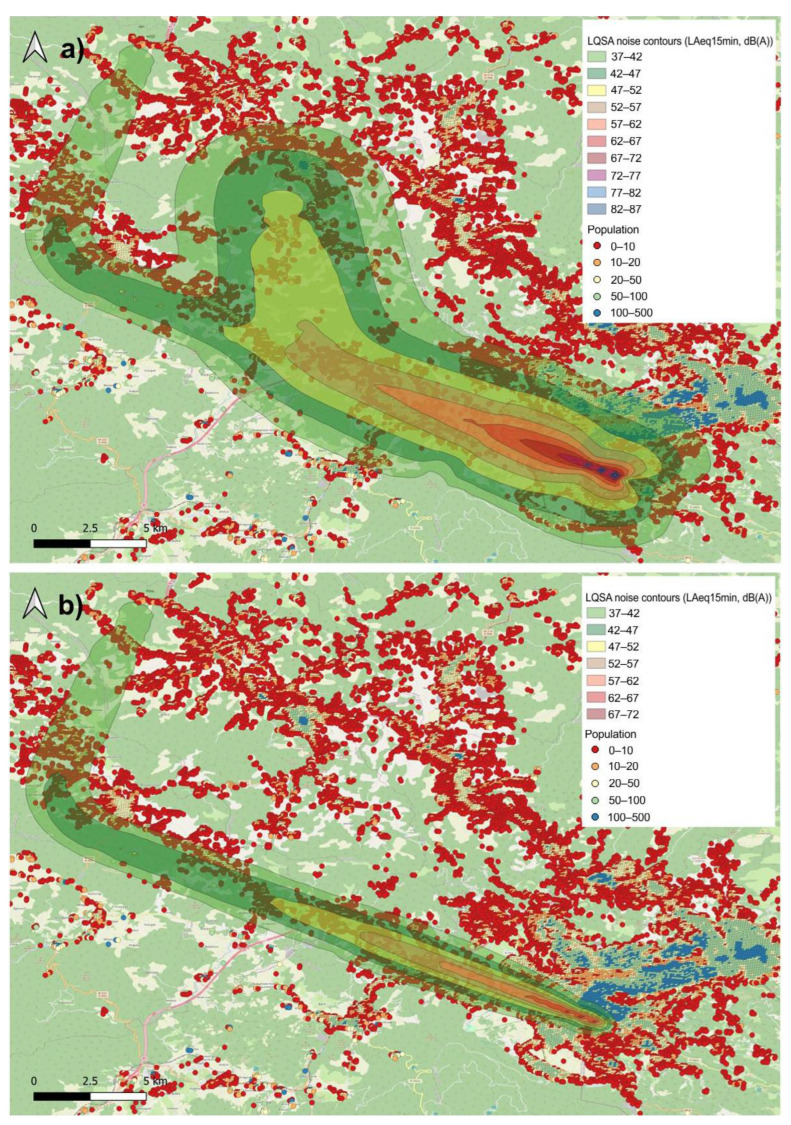
*L_Aeq_*_,15min_ noise contours for Sarajevo Airport, (**a**) with TAM-related delays and (**b**) without TAM-related delays.

**Figure 5 ijerph-19-08921-f005:**
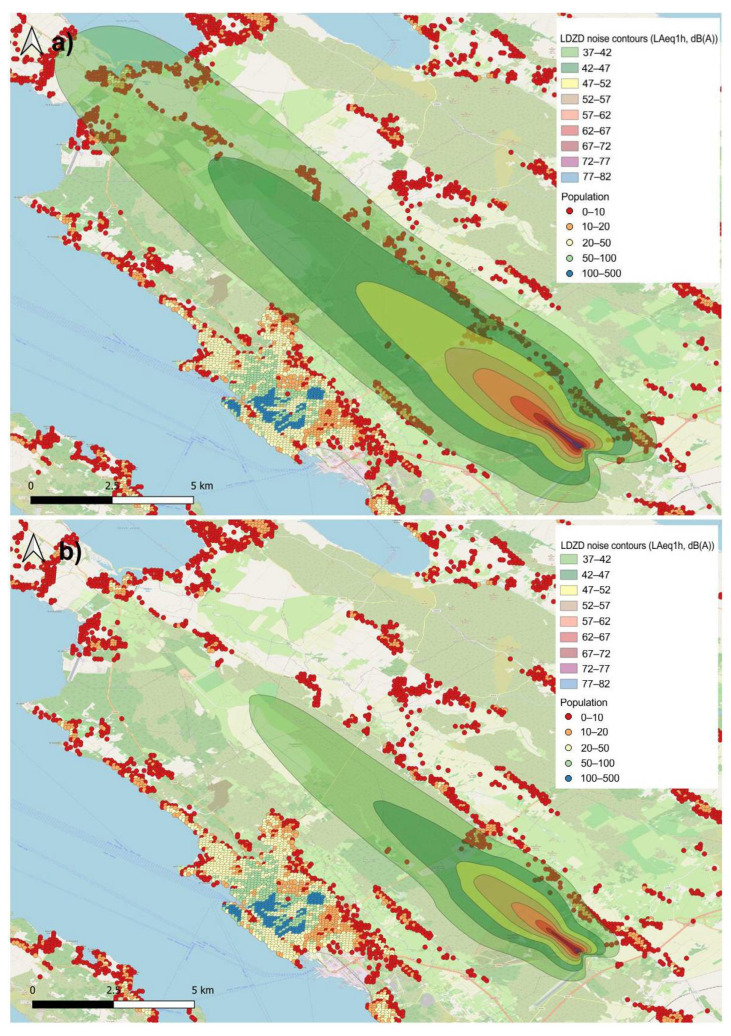
*L_Aeq_*_,1h_ noise contours for Zadar Airport, (**a**) with TAM-related delays and (**b**) without TAM-related delays.

**Figure 6 ijerph-19-08921-f006:**
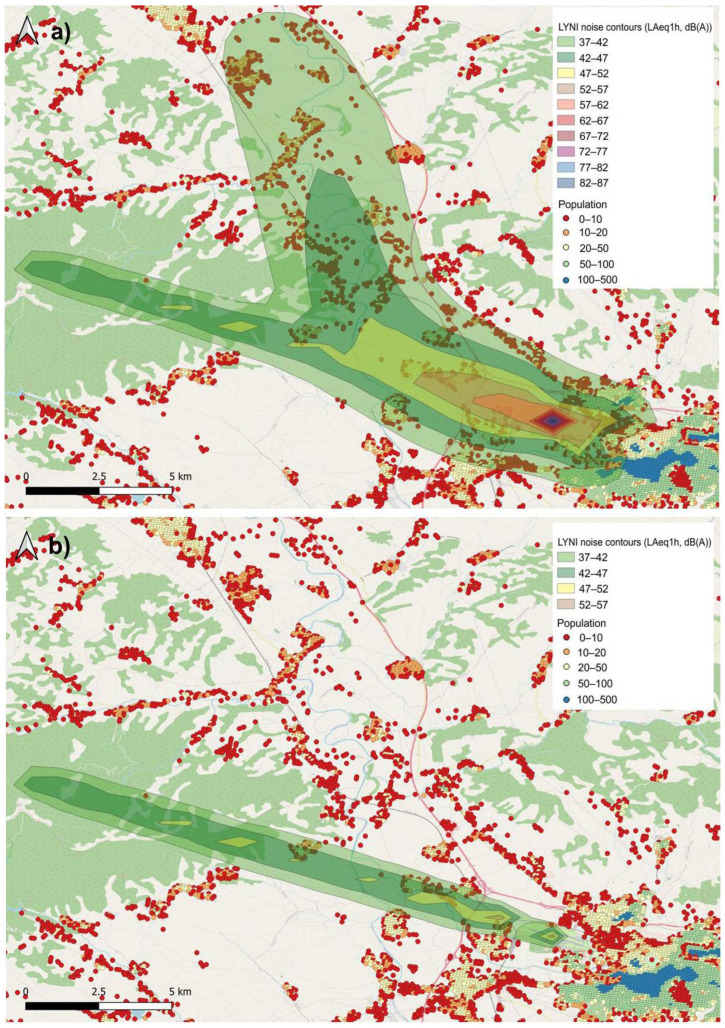
*L_Aeq_*_,1h_ noise contours for Niš Airport, (**a**) with TAM-related delays and (**b**) without TAM-related delays.

**Table 1 ijerph-19-08921-t001:** Number of people annoyed and highly annoyed around the three airports.

		Sarajevo Airport	Zadar Airport	Niš Airport
**Number of Intervals Analysed**		43	20	24
**Number of Annoyed People (Intervals Without TAM Delays)**	Minimum	585	0	201
	Maximum	45,401	604	16,631
	Average	20,583	229	2442
	Mean	21,018	183	772
	First Quartile (Q1)	12,433	29	343
	Third Quartile (Q3)	27,655	387	1540
**Number of Annoyed People (Intervals with TAM Delays)**	Minimum	9175	36	851
	Maximum	53,136	869	17,836
	Average	28,222	385	4316
	Mean	27,330	368	2813
	First Quartile (Q1)	17,476	243	2278
	Third Quartile (Q3)	36,965	512	3848
**Number of Highly Annoyed People (Intervals without TAM Delays)**	Minimum	53	0	24
	Maximum	16,262	116	3515
	Average	6438	26	441
	Mean	6344	8	120
	First Quartile (Q1)	3222	0	67
	Third Quartile (Q3)	8936	37	306
**Number of Highly Annoyed People (Intervals with TAM Delays)**	Minimum	2634	0	126
	Maximum	19,771	161	3735
	Average	9518	45	729
	Mean	9073	28	486
	First Quartile (Q1)	5340	12	388
	Third Quartile (Q3)	12,609	56	592

## Data Availability

The data regarding population in the airport’s surroundings presented in this study are openly available from the European Commission, Joint Research Centre (JRC) at http://data.europa.eu/89h/jrc-ghsl-ghs_pop_eurostat_europe_r2016a (accessed on 26 April 2022). Restrictions apply to the availability of data regarding aircraft delays and air traffic. Data were obtained from the airport authorities and are available only with their permission.

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
