# Peer review of "Impact of Aircraft Delays on Population Noise Exposure in Airport’s Surroundings"

_ijerph, 2022, doi:10.3390/ijerph19158921_

Round 1

Reviewer 1 Report

It is interesting to try to capture how aircraft delays affect noise exposure and noise annoyance through case studies of the three airports.

However, only the effect of TAM related delay is considered, and the effect of delay when TAM is not introduced is not investigated. This is the limitation of this research.

It is stated that the Lden changes with delay, but this study uses LAeq,T instead of Lden.

I don't know why you used LAeq,T instead of Lden as the noise exposure level.

Please add explanation.

I'm not sure if “without TAM” includes or does not include delay. Please describe it clearly.

It is well understood that the delay changes the amount of exposure depending on the interval, but it is unclear how much it affects the daily average and how much it affects Lden.

The noise annoyance assessment of NPA and NHPA is for one year.

Is the noise exposure used to calculate the NPA short-term, or is it a total of one year's worth divided by intervals?

Since the amount of noise exposure used is not clear, it is not possible to judge the correctness of the analysis results and conclusions.

Line 162

What is EDP?

Please describe its definition if it is usually.

Line 183-

The description of “LAeq,T” is not unified. Please unify with standard writing style.

Equations 2 and 3 are officially equations for calculating NPA and NPHA when Lden is used.

Therefore, it should be recognized that the value of the result calculated by LAeq has no absolute meaning, and it should be mentioned earlier that it is used for relative comparisons.

Line 232

Please indicate when the population data is.

Section 4.2

Is there any recorded data of delays at Zadar Airport like Sarajevo Airport in Section 4.1?

Section 4.3

Is there any recorded data of delays at Nis “Constantine the Great” Airport like Sarajevo Airport in Section 4.1?

Section 5.1 TAM delays analysis

Lines 302-306

I don't know how to calculate 43 isolated intervals (Sarajevo Airport).

Please add more explanations. How does it relate to the sentence of lines 241-245?

  1. 43 times (days)? / One year?

It may be easier to understand if there is a unit in the number of 43.

Figures 2 and 3

I don't know how the Y-axis values were calculated.

Did the calculation of the exposed population use all the data for the number of delays / day, or did it use the data for a specific day?

There is no explanation as to what LQSA, LYNI and LDZD in the figures refer to.

If the blue line is Zadar, is the sentence of lines 340-341 correct?

Lines 342-344

How to see the word of “relative difference”?

Please add a description.

Lines 344-345

Are they similar in whole or in part?

Fig.5

It is unknown where Runway 31 is, so it is not possible to judge the correctness of the resulting sentence.

Why don't you make the figure a little bigger?

Fig. 6

It is unknown where Runways 29 and 11 are.

Author Response

The authors greatly appreciate all the comments given by Reviewer 1. A point-by-point responses to the reviewer’s comments have been provided. Please see the attachment.

Reviewer 2 Report

The manuscript reported the noise annoyance on the neighboring communities near three airports in Europe with and without TAM (Total Airport Management). Some results were presented and discussed. However, some important details regarding this research are missing, and the methodologies applied are not justified. Hence some of the results are not convinced. Moreover, the technical content is also relatively thin as a full research paper.

  1. As compared to other work, the application of time intervals “15 mins” and “60 mins”to evaluate the population under noise annoyance is new. However, there is no discussion or reliable criteria to justify the reliability of using these two time intervals.
  2. The unit for LAeq,T should be dB(A).

  1. In figure 1, it mentioned about “Calculation of affected population”-“number of people sleep-disturbed and highly sleep-disturbed by noise” in the methodology, however there is no results or discussion on this part.

  1. In section 3.2, different delays codes are mentioned in lines 161-163, how these delays influence the noise level is unclear.

  1. In page 4, the last paragraph, the so-called “three criteria for selecting relevant time interval” is too general and unclear.

  1. In section 3.3, more details regarding the “FAA’s Integrated Noise Model (INM)” are necessary. For example, what are the inputs and outputs? What is the accuracy?

  1. In Section 3.4, the methodologies in calculating the ??? (number of people annoyed) and NPHA (number of people highly annoyed) are not justified. In the quoted reference European Commission [44], the LAeq,T,L is based on a day-evening-night level (called “DNL” in the reference). However, through the entire manuscript, the time interval that is used to calculate LAeq,T,L is 15 mins or 60 min. Hence, it is doubted that the calculated ??? and ??H? are reasonable.

  1. In Section 5.1,the authors declared that “After analysing AODB data for Sarajevo, Zadar, and Niš airports and filtering those operations with TAM related delays, the results showed a difference in the number of operations and schedule distribution of the aircraft operations during the day.” However, no tables or figures to show the difference, and how it will affect the analysis? Hence, it is not convinced to simply state that “15-minutes interval for the Sarajevo airport and 60-minutes interval for the Zadar and Niš airports are relevant”.

  1. In lines 321 to 329, pls check that the values of increment such as “1.86 times”, “1.53 times”,“3.03 times”, “2.14 times” and so on are not corresponding to the numbers in Table 1. Moreover, it should be stated as “4.73 times” rather than “4.73”. Grammars and typos are needed to be corrected.

  1. In figures 2 and 3, it is not understood what the abbreviations “LQSA”、“LYNI”、“LDZD” stand for.

  1. In line 411, the authors stated that “The conducted research and the model could be applied to any airport, with adjustment of research parameters for selecting appropriate time intervals.” However, from the manuscript, it is still unknown regarding the “selection of appropriate time intervals”.

  1. In line 415, “The example of Sarajevo Airport indicates that a significant effect can be obtained for airports located in the city or in the vicinity of largely populated areas.” What is the significant effect referring to?

Author Response

The authors greatly appreciate all the comments given by Reviewer 2. A point-by-point responses to the reviewer’s comments have been provided. Please see the attachment.

Reviewer 3 Report

The results of this work are based upon the supposition that (1) the noise models for delayed aircraft movements are appropriate and (2) the annoyance model referenced is also appropriate. Now, both models are, within themselves, well referenced and proven to be useful in aircraft modelling and annoyance separately. Here the authors have brought together two "models" together to establish somewhat loose conclusions about how residents "could" be annoyed where delays in aircraft movements are expected. The idea is certainly interesting and the application of their work would be referenced and of particular suitability for stakeholders in design and planning. Unfortunately, this work lacks in the appreciation of the residents awareness of aircraft delays or movements which is not independent of them being habituated to the continuous daily flights overhead. The work lacks qualitative data  (and measured noise data too) on the behaviour and effect of delays (without reference to noise) from residents as "control" to the annoyance noise data. 
Perhaps the authors could review how they word/phrase their research as a submission.

Author Response

The authors greatly appreciate all the comments given by Reviewer 3. A point-by-point responses to the reviewer’s comments have been provided. Please see the attachment.

Reviewer 4 Report

This manuscript is well-written and easy for the reader to understand in a topic that is not very well explored in the literature. The authors have done a great job explaining the motivation and background behind the paper. I only have a few minor comments for improvement –

1. Lines 91-92 – "Delays that occur … not necessarily correlated". It will be nice to have an example to demonstrate this statement.

2. Lines 147-152 – "In order to … and arrival routes". I agree that all the listed data are important. However, thrust is one of the most important parameters when modeling noise. How is thrust accounted for in this methodology? An explanation is needed.

3. Line 188 – A reference to the INM tool should be added.

4. Lines 241-245, Lines 269-270, Lines 286-287 – Please add a source for the data regarding operation numbers.

5. Figures 4, 5, 6 – The legend is quite small and difficult to read, please consider making it larger and clearer.

Author Response

The authors greatly appreciate all the comments given by Reviewer 4. A point-by-point responses to the reviewer’s comments have been provided. Please see the attachment.

Round 2

Reviewer 1 Report

I am glad that you answered my points and comments politely.

The unknown points have been cleared.

Please check only one point (description of LAeq, T).

Isn't "AeqT" a subscript?

Reviewer 3 Report

Much improved.